# Is Hyperuricemia, an Early-Onset Metabolic Disorder, Causally Associated with Cardiovascular Disease Events in Han Chinese?

**DOI:** 10.3390/jcm8081202

**Published:** 2019-08-12

**Authors:** Kuang-Mao Chiang, Yuh-Chyuan Tsay, Ta-Chou Vincent Ng, Hsin-Chou Yang, Yen-Tsung Huang, Chen-Hsin Chen, Wen-Harn Pan

**Affiliations:** 1Institute of Biomedical Sciences, Academia Sinica, Taipei 11529, Taiwan; 2Institute of Statistical Science, Academia Sinica, Taipei 11529, Taiwan; 3Graduate Institute of Epidemiology and Preventive Medicine, National Taiwan University, Taipei 10055, Taiwan

**Keywords:** Onset sequence study, GWAS, Mendelian randomization, serum uric acid, Cardiometabolic disease, Taiwan Biobank, CVDFACTS

## Abstract

Background: Serum uric acid (SUA) has gradually been recognized as a potential risk factor for cardiovascular disease (CVD). However, whether the relationship is causal remains controversial. Methods: We employed two methods to demonstrate the importance of SUA in CVD development. First, we examined the onset sequence of hyperuricemia in relation to five cardiometabolic (CM) diseases. Second, we conducted a Mendelian randomization (MR) study to causally infer the relationship between SUA and CVD. The information collected from the Cardiovascular Disease Risk Factors Two-Township Study (CVDFACTS) and Taiwan Biobank was used, respectively. Results: The onset sequence study showed that hyperuricemia and hypo-alpha-lipoproteinemia (low HDL-C) have earlier ages of onset than other CM diseases. For the MR analysis, the high weighted genetic risk score (WGRS) group had a significantly increased cumulative lifetime risk of CVD compared with the low WGRS group (OR = 1.62, (1.17−2.23), *P* = 0.003). Sensitivity analysis using the WGRS derived from other populations’ SUA-influential SNPs revealed similar results. Conclusions: We showed that hyperuricemia is an earlier-onset metabolic disorder than hypertension, hypertriglyceridemia, and diabetes mellitus, indicating that high SUA plays an upstream role in CM development. Moreover, our MR study results support the idea that hyperuricemia may play a causal role in CVD development. Further validation studies in more populations are needed.

## 1. Introduction

Cardiovascular disease (CVD) is one of the leading causes of death worldwide. An estimated 17.5 million people die from CVDs annually, which accounts for 31% of global deaths [1]. Serum uric acid (SUA), the end product of purine metabolism and a correlate of metabolic risk factors [2], recently attracted attention as an independent risk factor for CVD [3]. In vitro and in vivo studies have suggested that SUA contributes to endothelial dysfunction by inducing antiproliferative effects on the endothelium and impairing nitric oxide production. Proinflammatory and proliferative effects of soluble uric acid (UA) on vascular smooth muscle cells were described. In animal models of mild hyperuricemia, hypertension developed in association with intrarenal vascular disease. Possible adverse effects of UA on the vasculature were linked to chemokine and cytokine overexpression, renin–angiotensin system activation, and increased vascular C-reactive protein. Although evidence suggests that UA has a complex participatory role in hypertension and atherosclerosis pathogenesis, whether the relationship between SUA and CVD is causal remains controversial because no clinical trials have been conducted [4,5]. Positive relationships between SUA and CVD have been consistently observed in Asians [5,6,7], who are generally considered highly susceptible to metabolic risk [8]. Therefore, it is warranted to investigate the elevation CVD risk associated with hyperuricemia in Asian populations.

We employed two methods to demonstrate the importance of SUA in the pathogenesis of CVD development. Metabolic syndrome, defined by the coexistence of cardiometabolic (CM) components (abdominal obesity, hypertension, diabetes mellitus, hypertriglyceridemia, and hypo-alpha-lipoproteinemia (Low HDL-C)), is a known CVD risk factor. The clustering of hyperuricemia with these CM components was thoroughly documented [9]. We have previously presented the cohort in the Cardiovascular Disease Risk Factors Two-Township Study (CVDFACTS) to demonstrate the onset sequences of the five components [10]. In the first part of this current study, we employed the same strategy to demonstrate an even earlier age-at-onset of hyperuricemia in relation to the five CM components.

We subsequently performed a Mendelian randomization (MR) study, a nature-run randomization trial, to confirm the causal relationship between SUA and CVD. Although a randomized controlled trial (RCT) is considered as the gold standard for evaluating causality, an RCT with an SUA-lowering agent is costly and requires a long period to demonstrate the effect on CVD events. Moreover, urate-lowering medication may have other side effects that undermine the benefits of urate-lowering treatment. For the above reasons, MR may serve as a timely alternative [11,12,13].

Several studies have used the MR approach to examine the causality between SUA and CVD outcomes in Caucasian populations; however, the results have been inconsistent. Although two studies [14,15] have suggested that SUA is causally associated with cardiovascular death, sudden cardiac death, and coronary heart disease (CHD), several studies [2,16,17] have reported no association. The single nucleotide polymorphisms (SNPs) employed in these uric acid Mendelian randomization (UAMR) studies comprised mainly those reported in the literature. Population variation in susceptibility genes should be considered. An SUA genome-wide association study (GWAS) recently demonstrated that among the SUA-SNPs that have been identified in European populations, only few can be repeatedly found in Chinese/Asian populations [18].

Compared with Europeans, the Chinese population is more susceptible to metabolic risk [8]. Therefore, conducting an MR study on the relationship between SUA and CVD in such a metabolic disease-prone population is warranted. In this MR study, we first determined the Han Chinese-specific SUA-associated SNPs through two-stage GWAS from the Taiwan Han Chinese Biobank (TWB). These SNPs were subsequently used to construct, through linear combination, a weighted genetic risk score (WGRS) as the SUA long-term exposure dosage for evaluating the causality between SUA and CVD. Finally, we used a logistic accelerated failure time location–scale mixture regression model [19] to examine the relationship between CVD and the WGRS. Additionally, we conducted a sensitivity analysis to validate our findings, in which we employed the WGRS constructed from SNPs reported in the literature (WGRS_L_).

## 2. Materials and Methods

### 2.1. Ethics Statement

Written informed consent was obtained from each participant at his/her initial clinic visit. The studies have been approved by the Internal Review Board of Academia Sinica (Permit Number: AS-IRB-BM-07021 for CVDFACTS and AS-IRB02-104160 for TWB). All methods were performed in accordance with the relevant guidelines and regulations.

### 2.2. Onset Sequence Study

First, we performed an onset sequence study to clarify the sequential relationships among hyperuricemia and the five CM components. As in our previous analysis of the five CM components [10], the same data on the five components collected during 1989–1990, 1990–1993, 1994–1997, and during 2000–2002 from the CVDFACTS cohort, were analyzed along with additional longitudinal data of hyperuricemia defined as SUA ≥ 7.7 mg/dL for men and ≥ 6.6 mg/dL for women. The CVDFACTS is a community-based longitudinal study that has been conducted since 1989 in two townships in Taiwan, Chu-Dung and Pu-Tzu [20,21]. Adults who are aged over 20 years at baseline, free from metabolic diseases at entry and had at least one follow-up examination, were included in the analysis. The disease status for each of the six diseases (diabetes mellitus, hypertension, hypertriglyceridemia, abdominal obesity, hypo-alpha-lipoproteinemia, and hyperuricemia) was determined from the longitudinal examinations, and we analyzed only data from men and women with a thoroughly defined disease status for each disease. The procedures for participant recruitment, measurements, and diagnoses of the CM diseases in the CVDFACTS have been previously reported [10,20,21]. The diabetes mellitus, hypertension, hypertriglyceridemia, abdominal obesity, hypo-alpha-lipoproteinemia, and hyperuricemia cohorts comprised 3816, 3970, 4330 3580, 2628, and 4016 participants, respectively. 

Left truncation, entailed by different study entry ages of the recruited disease-free participants, is a pertinent issue in longitudinal studies with repeated measurements. Moreover, the ages at disease onset could not be determined exactly because it is comprised of either interval-censored data or right-censored data. Therefore, we used the Turnbull–Frydman estimator [22,23] to plot the step-function curves for the overall age-of-onset distribution and the conditional age-of-onset distribution of the susceptible participants. Because the tail probabilities of most sex-specific overall event time curves were less than 1, indicating a substantial proportion of non-susceptible participants for most diseases [10,19], we employed the logistic-generalized gamma distribution (GGD) location–scale mixture regression models in the MixtureRegLTIC R package to simultaneously estimate the sex-specific probability of the susceptibility and the conditional age of onset of the susceptible participants for each disease (see the Appendix A for details). These sophisticated novel statistical methods and packages [10,19] have been advocated to enhance the analysis and interpretation of longitudinal studies in epidemiology and medicine.

### 2.3. UA Mendelian Randomization Study

We subsequently employed an MR approach to investigate the causal relationship between SUA and CVD. The genetic and exposure information of 10,000 participants (5000 men and 5000 women) from the TWB [24] were used. Blood samples were drawn at the first interview. The SUA level was measured using an Architect i2000SR Analyzer (Abbott Diagnostics, Abbott Park, Chicago, IL, USA). Cholesterol, triglycerides, and plasma glucose were measured using a Hitachi LST 008 automatic analyzer (Hitachi, Tokyo, Japan). Details on the TWB can be found on its official website (https://taiwanview.twbiobank.org.tw/index). The WGRS derived from SUA-influential SNPs was used as an instrumental variable to examine the causal relationship between SUA and CVD.

#### 2.3.1. Han-Chinese SUA-SNP Selection with a Two-Stage GWAS

For the UAMR study, a two-stage SUA-GWAS was conducted to identify Han Chinese-specific SUA-SNPs. In the first discovery stage, the whole-genome SNP data of 7000 randomly selected participants were used. In the second confirmatory stage, another independent sample set of 3000 participants was used to validate the SNPs found in the first stage. Finally, we combined the 10,000 samples in a combination analysis.

The Affymetrix Axiom Genome-Wide Array for the TWB (TWB chip) containing 649,711 SNPs was used for the whole-genome genotyping. The TWB chip is a customized hybridization-based oligonucleotide array jointly developed by Affymetrix Inc. (Santa Clara, CA, USA) and the Taiwan Biobank.

Uncertain kinships were detected based on the identity-by-state scores implemented using the PLINK open-source software. The 10,000 samples released by the TWB are unrelated. With regard to heterogeneity, according to our previous study, Han Chinese in Taiwan differ drastically in genotypic information compared with Caucasians but are relatively homogeneous among the three major ethnic subgroups: Minnan, Hakka, and Mainlanders [25].

Information on the genotyping call rate (GCR), Hardy–Weinberg equilibrium (HWE), and minor allele frequency (MAF) were used to evaluate genotyping quality. SNPs were excluded if any of the following five conditions were met: (1) nonpolymorphic in both cases and controls, (2) GCR < 0.95, (3) MAF < 0.01, (4) the SNP deviated from the HWE with *P* < 0.001 and (5) SNPs located on the sex chromosomes (Chr). Although we did not impute the missing SNP calls, only the SNPs with a high call rate (>95%) were included. Finally, 612,408 SNPs (approximately 94.3% of the SNPs on the TWB chip) were analyzed.

The SUA value, a quantitative trait, was treated as the dependent variable in the GWAS. The genotype derived from each SNP was treated as an ordinal variable (i.e., 0, 1, or 2). A linear regression model was used to identify the SUA-SNPs adjusted for the following covariates: age, sex, and body mass index (BMI). To adjust for the population stratification and batch effects, principle components (PCs) 1 to 5 derived from a PC analysis were also adjusted as covariates in the linear regression model. Multiple testing correction was performed to minimize false positives by using the false discovery rate (FDR) method [26]. PLINK [27] was used for the analyses.

In the second confirmatory stage, another independent sample set of 3000 participants was used to validate the SNPs found in the first stage. The same genotyping platform and analysis methods were employed in the association tests at the second stage. Finally, in the combination analysis, we combined the 10,000 samples to complete the association test. The *β* values from the combination analysis were used to calculate the WGRS. 

Overall, 116 SNPs in nine loci were significantly associated with SUA. Gene regions and SNP positions were annotated using VarioWatch [28], which uses the National Center for Biotechnology Information’s *Homo sapiens* Annotation Release 105 and the Database of Single Nucleotide Polymorphisms (141) as references. Included in the WGRS calculation were the SNPs on eight previously reported SUA-related genes.

The correlations among the SNPs in each gene region were high. If all SNPs were included to generate the WGRS, the WGRS may have been dominated by larger genes with more SNPs in the linkage disequilibrium (LD) block. To account for such a potential bias, only single representative SNPs with the highest SUA effect size in each region were selected for calculating the WGRS. Appendix A presents the SNP selection. Haploview [29] was used to perform the LD analysis and to visualize the LD structures for the eight regions, as shown in Appendix A. Eight representative tag-SNPs were used to calculate the WGRS.

#### 2.3.2. SUA-SNPs from Literature Review

We constructed the WGRS by using the successfully replicated SUA-SNPs presented in the literature (WGRS_L_). Most of these successfully replicated SUA-SNPs were identified in European populations. A recently published SUA-GWAS of 12,281 Chinese participants demonstrated that among the 30 satisfactorily replicated SUA-SNPs identified in European populations, nine SNPs on six genes are replicated in the Chinese population [18]. Among these six genes, *HNF4G* and *IGF1R* are two widely known genes associated with diabetes mellitus, which we further eliminated. Consequently, we considered seven Han Chinese-specific SNPs from four genes in the sensitivity analysis. 

Because some SNPs are LDs, we decided to select only one SNP from each gene. Finally, four SNPs were used to construct the WGRS_L_. We used the data from 10,000 participants in the TWB to individually assess the association between these four SNPs and SUA levels. Age, sex, and five PCs (PC1 to PC5) for stratification and batch effects were adjusted. The obtained *β* values were subsequently used to construct the WGRS_L_ (Appendix A).

#### 2.3.3. Calculating the WGRS

We constructed the WGRS as a linear combination of the selected SNPs (0, 1, or 2) weighted by their *β* coefficients on SUA obtained from the linear regression models of the SUA-GWAS analysis: WGRS = *β*_1_ × SNP_1_ + *β*_2_ × SNP_2_ + … + *β*_8_ × SNP_8_. Because the *β* values of the selected SNPs were similar between the two independent sample sets (7000 participants in the first stage and 3000 participants in the second stage) (Table 1), we used the pooled *β* from the 10,000 joint samples to calculate the WGRS.

#### 2.3.4. Association between SUA and CVD Events

Information on self-reported CHD and stroke along with their age of onset was collected from the TWB; the CVD event time was defined by the age of onset of either CHD or stroke, whichever was earlier. We previously reported satisfactory validity for self-reported CVD status among community-dwelling Taiwanese using data from the CVDFACTS (see the Appendix A for the details).

The age of onset of CVD by the patients’ self-report and age at recruitment of non-patients was used as the right-censored event time in the association study of SUA-WGRS and CVD. To examine the causal relationship between SUA and CVD, the Turnbull–Frydman estimator [22,23] for the event time curve was used to plot the sex and the SUA-WGRS-specific overall age-of-onset distribution and the corresponding conditional age-of-onset distribution of susceptible participants. As used in the onset sequence study, the logistic-GGD location–scale mixture regression model [10,19] was also employed to examine the association between SUA-WGRS and CVD adjusted for sex. As exposure to SUA at a dosage was determined using the genotypes (SUA-WGRS) and started from birth, it is reasonable that the SUA-WGRS was used as the age-independent risk factor in the association study of CVD. 

### 2.4. UAMR Study: Sensitivity Analyses

To ensure that the selected SNPs directly affected SUA, which in turn affected the CVD outcome, we used MR-Egger regression to evaluate the potential pleiotropic effects. The intercept from the MR-Egger regression provided a test for directional pleiotropy. Additionally, the slope of the MR-Egger regression provided pleiotropy-corrected causal estimates; however, the estimated slope was underpowered unless the SNPs were combined to explain a large proportion of the variance in the exposure [30,31]. The MendelianRandomization R package was used to conduct the MR regression.

## 3. Results

### 3.1. Onset Sequence Study

Appendix A shows the optimally fitted regression model for each of the six CM diseases. Figure 1 illustrates the estimated overall event time curve (upper panel) and overall probability density curve (lower panel) and indicates the peak age of onset of each metabolic syndrome component in men (left panel) and in women (right panel). The Turnbull–Frydman sex-specific overall event time curves revealed full susceptibilities to hyperuricemia, abdominal obesity, and hypertension in both sexes and hypo-alpha-lipoproteinemia in men. By using the mixture model, the estimated probabilities of the susceptible men and women combined were 0.50 and 0.81 for diabetes mellitus and hypertriglyceridemia, respectively, and 0.70 for hypo-alpha-lipoproteinemia in women.

The overall density curves revealed that the onsets of hyperuricemia, hypolipoproteinemia, and abdominal obesity in women tended to occur in young adulthood, followed by hypertension and hypertriglyceridemia in middle age, and diabetes mellitus in later life (right panel of Figure 1b). For men, the onsets of hyperuricemia and hypo-alpha-lipoproteinemia tended to occur first, followed by abdominal obesity and hypertriglyceridemia in young adulthood, hypertension in middle age, and diabetes mellitus in later life (left panel of Figure 1b). 

Briefly, the CVDFACTS data presented an early age of onset for hyperuricemia and hypo-alpha-lipoproteinemia. However, the MR study on CVD events with respect to hypo-alpha-lipoproteinemia produced a null result (Appendix A); thus, we report below only the MR study findings on the causal inference for the relationship between SUA and CVD.

### 3.2. MR Study: SUA-SNP Discovery and Selection

In the first stage (*N* = 7000) of the two-stage GWAS, 147 SNPs were significantly associated with the SUA level (positive FDR < 0.05, Figure 2). In the second stage using another 3000 independent samples, 116 SNPs were confirmed. Ordered by the level of statistical significance, 108 SNPs are located on Chr 4, six SNPs on Chr 2, one SNP on Chr 1, and one SNP on Chr 7. These regions contain the following genes: *MUC1* (Chr 1), *GCKR* (Chr 2), *SLC2A9*, *PKD2*, *ABCG2*, *WDR1*, *IBSP*, and *MEPE* (Chr 4). The SNP on Chr 7 was not near any annotated gene regions. 

We compared these identified regions with those in the literature. Apart from one SNP on Chr 7, the other eight genes have been reported as SUA-related genes. To construct WGRS, eight representative tag-SNPs were selected, each from one of the eight confirmed gene regions. These SNPs were rs4072037 (on *MUC1*), rs1260326 (on *GCKR*), rs3733588 (on *SLC2A9*), rs2725211 (on *PKD2*), rs4148155 (on *ABCG2*), rs3756224 (on *WDR1*), rs17013187 (on *IBSP*), and rs17013282 (on *MEPE*) (see Table 1 and Appendix A for details of these eight representative SNPs, namely location, gene name, F-statistic, R-square, and *P* values for the GWAS).

We have also collected information on the SUA-SNPs from the literature. To construct the WGRS_L_, we used four satisfactorily replicated SNPs that were confirmed in the Chinese population but are unassociated with other metabolic traits. These SNPs are rs11264341 (on *TRIM46*, Chr 1), rs780094 (on *GCKR*, Chr 2), rs11722228 (on *SLC2A9*, Chr 4), and rs4148155 (on *ABCG2*, Chr 4) (Appendix A).

### 3.3. MR Study: WGRS

The F-statistic and R-square of the WGRS were 592.01 and 0.32, respectively, indicating a favorable relationship between SUA and the genetic instrumental variable score constructed from the set of SNPs selected from the TWB [32]. The sensitivity analysis of the literature-mined SNPs indicated a reasonably favorable F-statistic (372.99) and R-square (0.036) for the WGRS_L_. 

### 3.4. MR Study: Relationship between SUA and CVD

Of the 10,000 participants in the TWB, 179 patients showed CVD events and 9821 peers did not. As expected, the characteristics of patients with CVD and their peers without CVD differed significantly in terms of distributions of sex, age, UA, BMI, alcohol consumption, smoking status, education level, marital status, exercise status, fasting glucose (FG), total cholesterol (T-CHO), high- and low-density lipoprotein cholesterol (HDL-C/LDL-C), fasting triglyceride (TG), estimated glomerular filtration rate (eGFR) and systolic/diastolic blood pressure (SBP/DBP) (Appendix A). However, because MR is used to mimic an RCT, we examined whether people in four SUA exposure groups (Q1, Q2, Q3, and Q4 of the WGRS quartiles) were comparable. The characteristics of these groups are provided in Table 2 and Table 3 (Appendix A shows sex stratified results). As expected, mean SUA values increased incrementally as the WGRS continued. No significant differences were observed in the WGRS groups in sex, physical activity, alcohol consumption, smoking status, education level, marital status, exercise status distribution, mean values for age, BMI, FG, T-CHO, LDL-C, eGFR, SGOT, SGPT and SBP. A J-shape relationship for TG and a modest increase in DBP were observed.

The fitted logistic-GGD location–scale mixture regression models with sex adjustment are presented in Table 4 in which we treated the SUA-WGRS as either a quantitative variable or a group variable. For group comparisons, the WGRS was classified into four quartile groups (Q1, Q2, Q3, and Q4) or two groups (low and high, defined by the median). All three fitted mixture models indicated that the SUA-WGRS was significantly and positively associated with the CVD outcome only in the logistic regression part (see next paragraph). In other words, sex and WGRS did not affect the age of onset (see location and scale parts) of CVD for patients susceptible to CVD. When comparing the CVD cumulative lifetime risk between the sex and SUA-WGRS-specific groups, the resultant smooth overall event time curves from the fitted mixture model were close to the Turnbull–Frydman step-function event time curves shown in Figure 3.

Part (a) of Table 4 reveals that the cumulative lifetime risk (or susceptibility probability) of CVD increased with the quantitative level of the WGRS (*P* = 0.029). Table 4(b) and Figure 3a show that Q1 and Q2 could be reasonably merged into the low WGRS group, and Q3 and Q4 could be merged into the high WGRS group. Table 4(c) reveals that the high WGRS group had higher probabilities of being susceptible to CVD (4.82% for women and 12.60% for men) than the low WGRS group (3.03% for women and 8.17% for men) with an odds ratio (OR) of 1.620 (95% confidence interval (CI): 1.175–2.232, *P* = 0.003). The estimated overall event time curves and overall density curves of the high WGRS groups versus the low WGRS groups in Figure 3b provide corresponding visual displays for both women and men. 

We used the WGRS_L_ constructed from the four satisfactorily replicated Han Chinese SUA-SNPs gathered from the literature to validate the causality between SUA and CVD by employing the same model (Appendix A). Appendix A reveals that the high WGRS_L_ group also had a significantly higher CVD risk than the low WGRS_L_ group (OR = 1.47, 95% CI: 1.07–2.02, *P* = 0.018).

The results of the MR-Egger regression analysis are provided in Appendix A for examining the pleiotropic effects of these SNPs on the CVD outcome. The estimated intercepts from the MR-Egger regression for all 85 tag-SNPs or eight representative SNPs were both near zero (85 tag-SNPs: intercept = 0.006, *P* = 0.9; eight representative SNPs: intercept = 0.05, *P* = 0.63), revealing no significant pleiotropic effect. Similar results were observed for the previously reported SNPs. Appendix A presents a scatterplot of the genetic associations and causal estimates by using the 85 tag-SNPs.

We also conducted regression analyses for the CHD and stroke outcomes separately, comparing the risk of the two-group SUA-WGRS. The optimally fitted regression models in Appendix A and visual patterns in Appendix A indicate that, similar to CVD, the two-group SUA-WGRS was significantly and positively associated with the cumulative lifetime risk of CHD (*P* = 0.007) but not significantly associated with that of stroke. However, the positive coefficient estimate of the scale regression part for stroke (*P* = 0.023) demonstrated a significantly wider range of stroke age at onset for susceptible women and men in the high SUA-WGRS group (29–69 years old) than for those in the low SUA-WGRS group (39–62 years old).

## 4. Discussions

In this study, we demonstrated that hyperuricemia is an earlier-onset metabolic disorder compared to hypertriglyceridemia, hypertension, and diabetes mellitus, indicating that high SUA plays an upstream role in CM disease development. The results from the MR study, a nature-run randomized trial, support that hyperuricemia may play a causal role in CVD development.

In the MR study, we determined that the SUA-WGRS was significantly associated with an increased cumulative lifetime risk of CVD events, whereas MR-Egger regression analyses did not support the pleiotropic effect of these SNPs, i.e., effects on CVD not through hyperuricemia. This is the first MR study on Han Chinese to suggest long-term exposure (after birth) to a systematically higher SUA from genetic disposition contributes to an increased cumulative lifetime risk of CVD.

The age of onset of hypo-alpha-lipoproteinemia (low HDL-C) is also early and close to that of hyperuricemia in young adulthood. Thus, we also conducted an MR study on HDL-C and CVD by using the same method as a negative control (Appendix A). The WGRS of HDL-C was not associated with CVD (high vs. low: OR = 0.91, CI: 0.66–1.24, *P* = 0.54), indicating that HDL-C is not a causal factor for increased lifetime risk of CVD.

Congruent with our findings, a recent meta-analysis that pooled 10 cohort studies with data from 172,123 participants demonstrated that baseline SUA is an independent risk factor of CVD mortality [3]. Our earlier epidemiological study determined that SUA is independently associated with CVD risk not only in the general population [5] but also in those without conventional metabolic disorders [33]. Furthermore, our recent study analyzing insurance data [34] showed that those people on urate-lowering medication experienced a lower risk of CVD events compared with those who were not. However, due to a lack of results from large-scale clinical trials, the causal relationship between SUA and CVD disease has not been established [35,36]. Our study, which utilized a sophisticated statistical method of mixture regression models, supports the causal relationship between hyperuricemia and an increased cumulative lifetime risk of CVD previously observed in prospective investigations. Future meta-analyses on the MR findings are warranted.

Several MR studies have investigated the causal relationship between SUA and CVD in Caucasian populations but have produced mixed results [2,14,15,16,17]. Examining the causal relations between the SUA-WGRS and CVD outcomes in the Han Chinese population is warranted because East and South Asians have relatively high susceptibilities to metabolic diseases. Despite a relatively low BMI range among Han Chinese, the prevalence rates of type 2 diabetes and gout (10% [37] and 8.21% [38], respectively) are higher than those in the United Kingdom (6.2% [37] and 1.4% [39], respectively), for example. This may partly explain why hyperuricemia has been associated with CVD in most Chinese studies, including those carried out in Taiwan and our UAMR study. Small-scale studies have shown that urate-lowering medication reduces blood pressure in adolescents with hyperuricemia [40], improves exercise capacity in patients with chronic stable angina [41], and ameliorates endothelial function in patients with heart failure [42]. 

In the current study, the SUA-GWAS identified and confirmed 116 SNPs in eight SUA-associated genes. The functions of these eight genes are related to urate transport, osteoblast metabolism, and kidney function (see the Appendix A for details on the gene functions).

Although our study has suggested a possible causal role of UA in the CVD, the biological mechanism is still not clear. Several possible mechanisms have been proposed to explain the relationship between UA and CVD development. Monosodium urate crystals may be phagocytized by immune cells and may activate the nucleotide-binding oligomerization domain-like receptor protein 3 (NLRP3) inflammasome [43], which can secrete proinflammatory cytokines IL-1a and IL-1b and increases sarcoplasmic reticulum Ca^2+^ leakage, leading to depressed contractility and arrhythmia [44]. NLRP3 activation and proinflammatory cytokine secretion may promote atherosclerosis. 

Moreover, chronic hyperuricemia stimulates the renin-angiotensin system and inhibits endothelial nitric oxide (NO) release, which may lead to renal vasoconstriction and increases blood pressure. Persistent renal vasoconstriction would contribute to arteriolosclerosis and the development of hypertension and CVD [4]. Calmodulin (CAM) may be the possible link underlying this mechanism [45]. UA can bind directly to CAM and interfere with the binding of CAM to endothelial NO synthase. Missense mutations in CAM have been linked to certain inherited forms of catecholaminergic polymorphic ventricular tachycardia that greatly increases CVD risk.

Moreover, UA may function as a pro-oxidant when present in large amount. A functional study revealed that hyperuricemia induces redox-dependent signaling and oxidative stress in adipocytes [46]. Oxidative stress induced by hyperuricemia may in part explain the increment of CVD risk in hyperuricemia patients. Recent studies have demonstrated that blocking XOR also reduces oxidative stress [47]. Xanthine oxidoreductase inhibitors (XORIs) are often used as urate-lowering agents because xanthine is a product on the pathway of purine degradation and XORIs block the conversion of xanthine to UA by xanthine oxidase and in turn reduce both intra- and extracellular urate. Intracellular urate may also stimulate NADPH oxidase [47]. The NADPH oxidase family contributes to major sources of reactive oxygen species (ROS) that have been implicated in the pathophysiology of many cardiovascular diseases [48]. Many clinical trials have shown that treatment with XORIs can improve the HT, CKD, MetS, and insulin resistance.

The SUA effects on CVD risk may be related to non-alcoholic fatty liver disease (NAFLD), since NAFLD has also been recognized as an early onset clinical disorder which links to both MetS and CVD [49,50,51,52,53,54,55]. Moreover, NAFLD patients are often accompanied with hyperuricemia. Accumulating evidence has shown that the SUA level was an independent predictor of NAFLD. Besides, previous studies [56] have shown that the prevalence and severity of NAFLD are higher in men than in women in young adulthood and a crossover phenomenon between sexes was seen after menopause [57]. These sexual dimorphism phenomena in both traits may suggest a link between pathogenesis of hyperuricemia and NAFLD. The link between SUA, NAFLD and CVD is apparently more complicated than previously believed. Although our MR study did not find association between SUA genetic score and markers of liver function (Table 2), the interrelationship of UA, NAFLD and CVD is worthy of further in-depth investigation.

Besides, our study has some limitations. First, the CVD outcome is self-reported and includes both heart disease and stroke. However, our validation study with the CVDFACTS data demonstrated reasonable consistency (90%) between the self-reported CVD events and National Health Insurance data. We conducted separate analyses for heart disease and stroke. The resultant sex-specific overall event time curves and overall probability density curves in Appendix A demonstrate that consistent with CVD, the high SUA-WGRS group had a significantly higher cumulative lifetime risk of CHD than did the low SUA-WGRS group for both men and women, but power was insufficient for this phenomenon to be demonstrated for stroke. Nonetheless, susceptible women and men in the high SUA-WGRS group had a wider range of stroke ages-at-onset than those in the low SUA-WGRS group. These results may provide a future direction for etiological study of CVDs. Second, some of the identified genes may have a pleiotropic effect. For example, *MUC1*, *SLC2A9*, *GCKR*, and *PKD2* may influence other metabolic syndrome–related traits such as blood pressure and FG. However, no significant pleiotropic effects were identified using the MR-Egger regression approach. To further verify our results, we removed the four SNPs located on the aforementioned potential pleiotropic genes to test the associations between the new SUA-WGRS and CVD. This sensitivity analysis (Appendix A) produced a similar result because the high WGRS group still had a significantly increased cumulative lifetime risk of CVD compared with the low WGRS group (high vs. low: OR = 1.58, 95% CI: 1.15–2.18, *P* = 0.005) for both sexes. We also conducted a sensitivity analysis which used the satisfactorily validated SUA-SNPs of the Chinese population extracted from the literature. Although the significance level was reduced, the results supported our findings, indicating that our study is unlikely to suffer from the winner’s curse.

In conclusion, we determined that the peak age of onset of hyperuricemia is in early young adulthood. Furthermore, we discovered eight Han Chinese SUA-associated genes and representative SNPs. The MR results, either using SNPs obtained from the TWB or those reported in the literature, revealed that genetically induced SUA increments may causally increase the cumulative lifetime risk of CVD in Han Chinese.

Data Availability: The data we used can be applied from the Taiwan Biobank at https://www.twbiobank.org.tw/new_web_en/about-export.php.

## Figures and Tables

**Figure 1 jcm-08-01202-f001:**
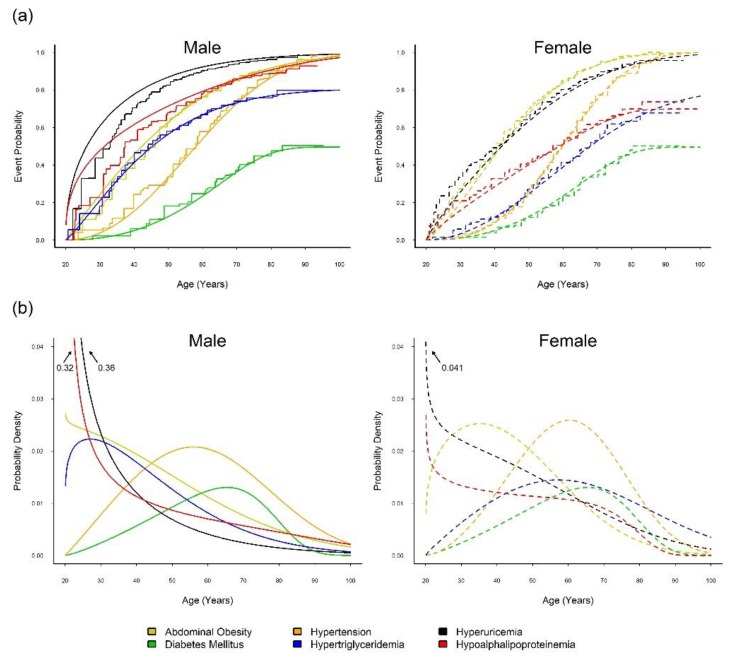
Time-to-event curves and probability density curves of five metabolic syndrome (MetS) components and hyperuricemia. (**a**) Smooth and step event time curves, and (**b**) probability density curves, of five MetS components and hyperuricemia for male (solid line) and female (dashed line), estimated from the optimally fitted models of the Cardiovascular Disease Risk Factors Two-Township Study (CVDFACTS) in Taiwan, 1989–2002.

**Figure 2 jcm-08-01202-f002:**
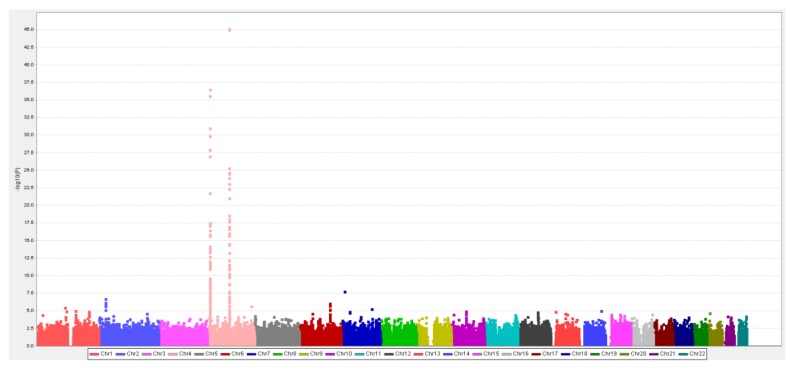
First-stage serum uric acid (SUA)-GWAS results: age, sex, BMI, and five PCs (concerning stratification and batch effects) were adjusted in the linear regression model.

**Figure 3 jcm-08-01202-f003:**
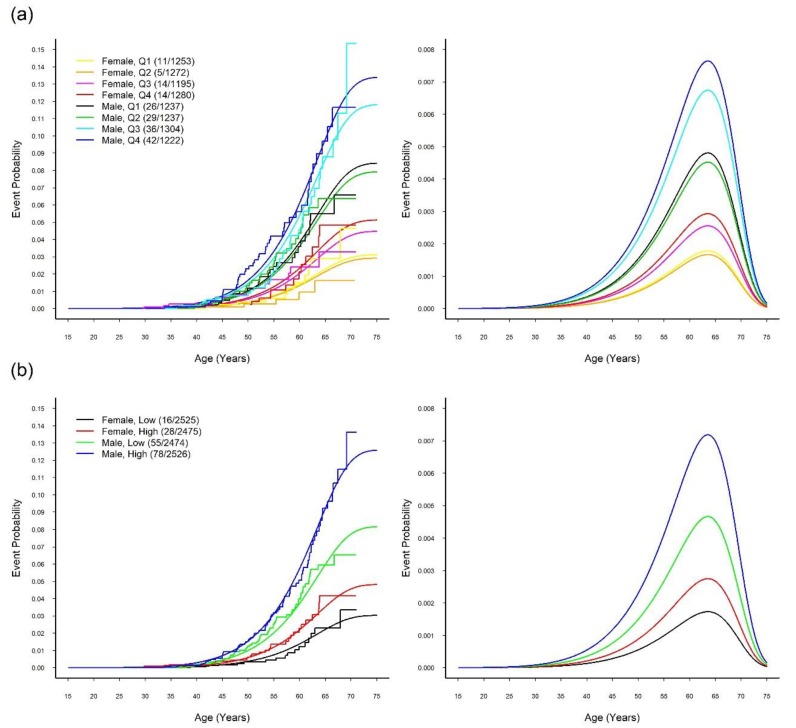
Time-to-event curves and probability density curves of CVD event. Smooth and step event time curves (left panel) and probability density curves (right panel) estimated from the optimally fitted logistic-GGD location–scale mixture regression models of the CVD disease stratified by (**a**) sex and four SUA-WGRS groups and (**b**) sex and two SUA-WGRS groups.

**Table 1 jcm-08-01202-t001:** Single nucleotide polymorphisms (SNPs) selected from two-stage genome-wide association study (GWAS).

					1st Stage (*N* = 7000)			2nd Stage (*N* = 3000)		Combined (*N* = 10,000)			
Chr	SNP	BP	Gene	Alleles	Beta	*P*-Value	FDR	Beta	*P*-Value	Beta	*P*-Value	F	R^2^
4	rs4148155	89054667	*ABCG2*	A/G	0.31	9.35 × 10^−46^	2.86 × 10^−40^	0.32	5.60 × 10^−22^	0.31	4.34 × 10^−66^	37.7	0.022
4	rs3733588	9997303	*SLC2A9*	A/G	−0.26	2.91 × 10^−37^	5.93 × 10^−32^	−0.21	2.80 × 10^−11^	−0.24	1.73 × 10^−46^	25.9	0.015
4	rs2725211	88970375	*PKD2*	C/T	0.25	1.86 × 10^−25^	1.14 × 10^−20^	0.30	1.50 × 10^−17^	0.27	3.18 × 10^−41^	24.2	0.014
4	rs17013282	88765873	*MEPE*	G/A	0.19	1.03 × 10^−11^	1.15 × 10^−7^	0.21	1.38 × 10^−6^	0.20	4.95 × 10^−17^	11.1	0.007
4	rs17013187	88733531	*IBSP*	C/T	0.16	1.49 × 10^−9^	1.35 × 10^−5^	0.14	6.63 × 10^−4^	0.15	2.85 × 10^−12^	8.8	0.005
4	rs3756224	10105739	*WDR1*	T/C	0.11	1.05 × 10^−7^	6.67 × 10^−4^	0.10	1.73 × 10^−3^	0.11	9.45 × 10^−10^	7.9	0.005
2	rs1260326	27730940	*GCKR*	C/T	0.10	7.90 × 10^−7^	4.38 × 10^−3^	0.11	6.55 × 10^−4^	0.10	2.33 × 10^−9^	8.5	0.005
1	rs4072037	155162067	*MUC1*	T/C	0.11	8.18 × 10^−6^	3.60 × 10^−2^	0.11	4.56 × 10^−3^	0.11	1.43 × 10^−7^	6.8	0.004

GWAS with age, sex, BMI, and principle component (PC)1–5 (concerning stratification and batch effects) adjustment. Chr: Chromosome, BP: base pair, Alleles: Major allele/minor allele, FDR: *P*-value with false discovery rate (FDR) correction, F: F-statistical value.

**Table 2 jcm-08-01202-t002:** Sample characteristics by weighted genetic risk score (WGRS) quartile.

Characteristic	WGRS								*P*-Value ^†^
	Q1 (*N* = 2490)		Q2 (*N* = 2509)		Q3 (*N* = 2499)		Q4 (*N* = 2502)		
	Mean	(SD)	Mean	(SD)	Mean	(SD)	Mean	(SD)	
SUA (mg/dL)	5.4	(1.3)	5.6	(1.5)	5.8	(1.5)	6.0	(1.6)	<0.0001 *
Sex (Men%)	49.7%		49.3%		52.2%		48.8%		0.080
Age (yr)	49.1	(11.4)	48.9	(11.0)	48.6	(11.0)	48.8	(11.1)	0.140
BMI (kg/m^2^)	24.2	(3.6)	24.3	(3.6)	24.3	(3.6)	24.3	(3.6)	0.570
FG (mg/dL)	96.6	(21.6)	96.0	(20.7)	96.4	(19.1)	97.4	(25.0)	0.136
T-CHO (mg/dL)	193.4	(35.9)	191.8	(35.1)	191.9	(34.1)	194.1	(36.6)	0.060
TG (mg/dL)	118.0	(104.0)	113.8	(83.0)	117.9	(81.9)	124.7	(118.3)	0.002 *
HDL-C (mg/dL)	53.7	(13.4)	53.3	(13.2)	52.8	(12.8)	52.8	(13.1)	0.050
LDL-C (mg/dL)	120.3	(31.8)	120.4	(32.2)	120.4	(31.1)	121.4	(31.1)	0.520
eGFR (mL/min/1.73m^2^)	103.4	(24.4)	102.7	(24.6)	102.2	(24.7)	102.9	(25.1)	0.41
SGOT (U/L)	24.6	(16.6)	24.6	(12.5)	24.7	(13.1)	24.2	(10.4)	0.582
SGPT (U/L)	24.5	(18.8)	25.2	(21.6)	25.8	(23.5)	24.8	(19.4)	0.110
SBP (mmHg)	115.7	(16.9)	116.0	(17.1)	116.7	(17.3)	116.9	(18.0)	0.060
DBP (mmHg)	71.6	(10.8)	72.0	(10.9)	72.7	(11.3)	72.8	(11.3)	0.010 *

SUA: serum uric acid; BMI: body mass index; FG: fasting glucose; T-CHO: total cholesterol; TG: triglyceride; HDL-C/LDL-C: high-/low-density lipoprotein cholesterol; estimated Glomerular filtration rate; SGOT: serum glutamic-oxalocetic Transaminase; SGPT: serum Glutamic-Pyruvic Transaminase; SBP/DBP: systolic/diastolic blood pressure. *: *P* < 0.05. †: Chi-square test for categorical variables. Analysis of variance for continuous variables. Variables with skewed distribution were log-transformed before input for analysis. eGFR (mL/min/1.73 m^2^) = 175 × (Scr)^−1.154^ × (Age)^−0.203^ × (0.742 if women) × (1.212 if African-American).

**Table 3 jcm-08-01202-t003:** Sample characteristics by WGRS quartile (categorical traits).

Characteristic	WGRS								*P*-Value ^†^
	Q1 (*N* = 2490)		Q2 (*N* = 2509)		Q3 (*N* = 2499)		Q4 (*N* = 2502)		
	N	%	N	%	N	%	N	%	
**Sex**									0.08
Male	1237	49.7	1237	49.3	1304	52.2	1222	48.8	
Female	1253	50.3	1272	50.7	1195	47.8	1280	51.2	
**Drinking habit**									0.38
Yes	207	8.3	183	7.3	204	8.2	204	8.2	
No	2194	88.1	2239	89.2	2215	88.6	2232	89.2	
Quit	89	3.57	87	3.5	80	3.2	66	2.6	
**Smoking habit**									0.16
Yes	263	10.6	289	11.5	312	12.5	295	11.8	
Few	214	8.6	206	8.2	207	8.3	213	8.5	
No	1748	70.2	1723	68.7	1656	66.3	1703	68.1	
Quit	265	10.6	291	11.6	324	13.0	291	11.6	
**Education**									0.071
Elementary School	177	7.1	172	6.9	163	6.5	199	8.0	
Junior-high/Senior-high	1063	42.7	1045	41.7	984	39.4	1027	41.1	
BS/MS/PhD	1248	50.2	1289	51.4	1351	54.1	1276	51.0	
**Marriage**									0.66
Single	300	12.1	279	11.1	298	11.9	274	11.0	
Married	1936	77.8	1989	79.4	1961	78.5	1963	78.6	
Divorced/Widowed	253	10.2	238	9.5	239	9.6	262	10.5	
**Regular Exercise**									0.94
No	1453	58.4	1473	58.7	1472	58.9	1481	59.2	
Yes	1021	40.8	1027	41.1	1036	41.3	1037	41.7	

*: *P* < 0.05. †: Chi-square test for categorical variables.

**Table 4 jcm-08-01202-t004:** Optimally fitted logistic-GGD location–scale mixture regression models relating cardiovascular disease (CVD) events to sex and the WGRS.

Predictors	Logistic		Location		Scale		Shape		AIC
	OR (95% C)	*P*-Value	EST (95% CI)	*P*-Value	EST (95% CI)	*P*-Value	EST (95% CI)	*P*-Value	
**(a)** **continuous** **WGRS**									
Intercept	1	Referent	4.16 ^a^ (4.11, 4.21)	<0.001	−2.37 ^a^ (−3.00, −1.73)	<0.001	1.42 ^a^ (0.35, 2.49)	0.009	2594.09
Male	2.87 ^a^ (2.01, 4.10)	<0.001							
WGRS	1.41 ^c^ (1.04, 1.91)	0.029							
**(b)** **four-group WGRS**									
Intercept	1	Referent	4.16 ^a^ (4.11, 4.21)	<0.001	−2.37 ^a^ (−3.00, −1.73)	<0.001	1.42 ^b^ (0.35, 2.50)	0.009	2593.36
Male	2.86 ^a^ (2.00, 4.10)	<0.001							
Q2	0.94 (0.57, 1.53)	0.790							
Q3	1.46 (0.93, 2.29)	0.103							
Q4	1.68 ^c^ (1.08, 2.62)	0.022							
**(c)** **two-group WGRS**									
Intercept	1	Referent	4.16 ^a^ (4.11, 4.21)	<0.001	−2.37 ^a^ (−3.00, −1.73)	<0.001	1.42 ^b^ (0.35, 2.50)	0.009	2589.89
Male	2.85 ^a^ (1.99, 4.07)	<0.001							
High	1.62 ^b^ (1.17, 2.23)	0.003							

AIC: Akaike information criterion, CI: confidence interval, EST: estimated regression coefficient, OR: estimated odds ratio, WGRS: weighted genetic risk score. ^a^
*P* < 0.001, ^b^
*P* < 0.01, and ^c^
*P* < 0.05 for 2-sided *P*-value of the Wald test.

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
