# Peer review of "Is Hyperuricemia, an Early-Onset Metabolic Disorder, Causally Associated with Cardiovascular Disease Events in Han Chinese?"

_jcm, 2019, doi:10.3390/jcm8081202_

Round 1

Reviewer 1 Report

To elucidate the causal relationship of serum uric acid (SUA) in cardiovascular disease (CVD) development, the authors examined the onset sequence of hyperuricemia in relation to five cardiometabolic (CM) diseases and conducted a Mendelian randomization (MR) study to causally infer the SUA–CVD relationship using the information collected from the CVDFACT Study and Taiwan Biobank. The onset sequence study showed that hyperuricemia had earlier ages at onset than did other CM diseases. MR study showed that the high weighted genetic risk score (WGRS) group had a significantly increased cumulative lifetime risk of CVD compared with the low WGRS group. According to these findings, the authors concluded that high SUA plays an upstream role in cardiometabolic development and hyperuricemia has a causal role in CVD development.

The theme of this study is intriguing, the analyses were well-done, and the manuscript is well-written.

1. Sample characteristics are shown by WGRS quartile in Table 2-1. I suggest the authors to show them by WGRS and sex in the supplementary table.

2. How many participants developed CVD events in this study? The authors showed 179 participants in Table S3-1, while 177 participants in Table S7.

Author Response

Response to Reviewer 1 Comments

1. Sample characteristics are shown by WGRS quartile in Table 2-1. I suggest the authors to show them by WGRS and sex in the supplementary table.

Responses:

Thank you for the suggestion, we have added tables comparing men and women (table S4-1 and S4-2) in the supplementary.

Results (line 271)

The characteristics of these groups are provided in Table 2-1 and Table 2-2 (Table S4-1 and S4-2 shows sex stratified results).

2. How many participants developed CVD events in this study? The authors showed 179 participants in Table S3-1, while 177 participants in Table S7.

Responses:

Thank you for pointing out the mistake. In this study, a total of 179 patients developed CVD. We have correct the number in the Table S7. 

Reviewer 2 Report

From the methodological point of view this experimental study seems adequately appropriate; the study design process is valid, materials, methods, data and statistical analysis seem suitable to scientific rigour and ethical principles; results, figures and tables are appropriate and clear; References are updated and congruent.

I only ask the Authors to improve English and rectify some typing mistakes.

Author Response

Response to Reviewer 2 Comments

From the methodological point of view this experimental study seems adequately appropriate; the study design process is valid, materials, methods, data and statistical analysis seem suitable to scientific rigour and ethical principles; results, figures and tables are appropriate and clear; References are updated and congruent.

I only ask the Authors to improve English and rectify some typing mistakes.

Responses:

Thank you for your suggestion. We have reedited the English language by an English editor.

Reviewer 3 Report

The authors of the paper demonstrated an interesting thesis on a pathology of great interest. Furthermore, as already demonstrated by their previous published paper, they are applying solid medical statistics techniques to analyze the Cardiovascular Disease Risk Factors Two-Township Study (CVDFACTS). CVDFACTSis a community-based longitudinal study designed to investigate the development and evolution of cardiovascular diseasesin 2 townships of Taiwan. A further suggestion could be the comparison also with pathologies related to oxidative stress and levels in xanthine and xanthine oxidase serum that are two markers related to uric acid levels.

Author Response

Response to Reviewer 3 Comments

The authors of the paper demonstrated an interesting thesis on a pathology of great interest. Furthermore, as already demonstrated by their previous published paper, they are applying solid medical statistics techniques to analyze the Cardiovascular Disease Risk Factors Two-Township Study (CVDFACTS). CVDFACTS is a community-based longitudinal study designed to investigate the development and evolution of cardiovascular diseases in 2 townships of Taiwan. A further suggestion could be the comparison also with pathologies related to oxidative stress and levels in xanthine and xanthine oxidase serum that are two markers related to uric acid levels.

Responses:

Thank you for the suggestion. Since the oxidative stress and the levels in xanthine and xanthine oxidase were not measured in the CVDFACTS and TWB, we cannot provide the results of these markers in this study. Nevertheless, we have added a paragraph in the discussion on the relationships between these markers, oxidative stress, UA and CVD.

Discussion (line 371)

Moreover, UA may function as a pro-oxidant when present in large amount. A functional study revealed that hyperuricemia induces redox-dependent signaling and oxidative stress in adipocytes [46]. Oxidative stress induced by hyperuricemia may in part explain the increment of CVD risk in hyperuricemia patients. Recent studies have demonstrated that blocking XOR also reduces oxidative stress [47]. Xanthine oxidoreductase inhibitors (XORIs) are often used as urate-lowering agents because xanthine is a product on the pathway of purine degradation and XORIs block the conversion of xanthine to UA by xanthine oxidase and in turn reduce both intra- and extracellular urate. Intracellular urate may also stimulate NADPH oxidase [47]. NADPH oxidase family contributes to major sources of reactive oxygen species (ROS) that have been implicated in the pathophysiology of many cardiovascular diseases [48]. Many clinical trials have shown that treatment with XORIs can improve the HT, CKD, MetS, and Insulin resistance.

Reviewer 4 Report

GENERAL COMMENT

This study uses two different methods to evaluate the importance of SUA in  the development of CVD. First, the chronological primacy of hyperuricemia over other competing cardiometabolic diseases was assessed. Second, a Mendelian randomization study was conducted to further evaluate the association of SUA with CVD.

The topic is  not particularly novel although it retains paramount importance. I would suggest downplaying certain statements regarding the cause-and-effect relationship as opposed to a simple statistical association. Moreover, I would suggest reworking the discussion.

SPECIFIC COMMENT

Title and throughout the manuscript, try to more clearly define that statistical association does not prove causality. This is not a randomized controlled trial showing that the reduction of SUA will result in a decreased cardiovascular activity: this would be a more appropriate model to show causality. Therefore sentences such as  “Hyperuricemia is an early-onset metabolic disorder  causally associated with cardiovascular disease 3 events in Han Chinese”; “ However, whether the relationship is causal remains controversial.”; “ MR study results showed the causal role of hyperuricemia in CVD development. “ and so on appear to be inappropriate and must be reworked.

The background and the discussion must argue that NAFLD, which is an early occurrence/predictor of Metabolic Syndrome (Hepatology. 2005;41:64-71; Dig Liver Dis. 2010;42:320-30; Lancet Diabetes Endocrinol. 2014;2:901-10; Dig Liver Dis. 2015;47:181-90; J Gastroenterol Hepatol. 2016;31:936-44; J Hepatol. 2018;68:335-352), has recently been reported to be an independent cardiovascular risk factor  (J Hepatol. 2016 Sep;65(3):589-600) is also associated with SUA (Hepatol Res. 2016 ;46:1074-1087; World J Gastroenterol. 2017;23:6571-6592).  This could be, in theory, the link/one of the links putting together SUA with CVR.  Authors may be willing to develop this notion further.

Discussion – expand the paragraph discussing the limitations of the present study. Among these, please address that the present study offers no hints as to the putative biological mechanisms linking elevated SUA and cardiovascular events.  In particular, failure to evaluate the liver in these subjects must be acknowledged and extensively discussed.

Both NAFLD and SUA exhibit a definite sexual dimorphism. Comment on this.

This study uses sophisticated and somewhat uncommon statistical techniques. Make sure that these techniques are fully explained in clear terms so as to be rendered fully accessible also to clinicians who are inexperienced in this area.  In particular, describe strengths and limitations of these statistical techniques.

Author Response

Title and throughout the manuscript, try to more clearly define that statistical association does not prove causality. This is not a randomized controlled trial showing that the reduction of SUA will result in a decreased cardiovascular activity: this would be a more appropriate model to show causality. Therefore sentences such as “Hyperuricemia is an early-onset metabolic disorder causally associated with cardiovascular disease events in Han Chinese”; “ However, whether the relationship is causal remains controversial.”; “ MR study results showed the causal role of hyperuricemia in CVD development. “ and so on appear to be inappropriate and must be reworked.

Responses:

As we described in the background of the manuscript, although randomized controlled trials (RCTs) remain the gold-standard study design for inferring causality, they are exceedingly expensive and time-consuming efforts with high failure rates. In addition, RCTs are not always feasible or ethical to conduct. Mendelian Randomization (MR), is a well-established nature-run RCT which uses randomized variation in genes (SNPs) obtained through independent assortment in meiosis to examine the causal effect of an exposure determined by genetics on disease outcomes [1]. In recent years, hundreds of MR studies for various traits have been published to infer the causal relationship between the exposure variables and traits [2].

In the first part of the current study, we used the cohort in the Cardiovascular Disease Risk Factors Two-Township Study (CVDFACTS) to demonstrate the onset sequences of the five components of Mets together with uric acid. We further used MR to infer the causal relationship between the UA and CVD.

Since more research is needed to prove this causal relationship, we have modified the title and the sentences to make the inference more conservative.

Title:

“Is hyperuricemia, an early-onset metabolic disorder, causally associated with cardiovascular disease events in Han Chinese?”

Abstract (line 29):

MR study results support that hyperuricemia may play a causal role in CVD development. Further validation studies in more populations are needed.

Discussion (line 318):

The results from the MR study, a natural-run randomized trial, support that hyperuricemia may play a causal role in CVD development.

Conclusion (line 416):
The MR results, either using SNPs obtained from the TWB or those reported in the literature, revealed that genetically-induced SUA increments may causally increase the cumulative lifetime risk of CVD in Han Chinese

The background and the discussion must argue that NAFLD, which is an early occurrence/predictor of Metabolic Syndrome (Hepatology. 2005;41:64-71; Dig Liver Dis. 2010;42:320-30; Lancet Diabetes Endocrinol. 2014;2:901-10; Dig Liver Dis. 2015;47:181-90; J Gastroenterol Hepatol. 2016;31:936-44; J Hepatol. 2018;68:335-352), has recently been reported to be an ndependent cardiovascular risk factor (J Hepatol. 2016 Sep;65(3):589-600) is also associated with SUA (Hepatol Res. 2016 ;46:1074-1087; World J Gastroenterol. 2017;23:6571-6592). This could be, in theory, the link/one of the links putting together SUA with CVR.  Authors may be willing to develop this notion further

Responses:

Thank you for the suggestion. We have added a paragraph to address NAFLD in the discussion.

Discussion (line 382):

The SUA effects on CVD risk may be related to non-alcoholic fatty liver disease (NAFLD), since NAFLD has been also recognized as an early onset clinical disorder which links to both MetS and CVD [49-55]. Moreover, NAFLD patients are often accompanied with hyperuricemia. Accumulating evidences have shown that the SUA level was an independent predictor of NAFLD. Besides, previous studies [56] have shown that the prevalence and severity of NAFLD are higher in men than in women in young adulthood and a crossover phenomenon between sexes was seen after menopause [57]. These sexual dimorphism phenomena in both traits may suggest a link between pathogenesis of hyperuricemia and NAFLD. The link between SUA, NAFLD and CVD is apparently more complicated than previously believed. Although our MR study did not find association between SUA genetic score and markers of liver function (Table 2-1), the interrelationship of UA, NAFLD and CVD is worthy of further in-depth investigation.

Discussion – expand the paragraph discussing the limitations of the present study. Among these, please address that the present study offers no hints as to the putative biological mechanisms linking elevated SUA and cardiovascular events. In particular, failure to evaluate the liver in these subjects must be acknowledged and extensively discussed.

Responses:

Thank you for your suggestion. We have added the SGOT and SGPT in the Table 2-1. These markers of liver function were not significantly associated with SUA-WGRS since the MR study is in general not affected by the confounders. Besides, we have also expanded the limitation in the discussion to address these issues.

Results (line 276)

As expected, mean SUA values increased incrementally as the WGRS continued. No significant differences were observed in the WGRS groups in sex, physical activity, alcohol consumption, smoking status, education level, marital status, exercise status distribution, mean values for age, BMI, FG, T-CHO, LDL-C, eGFR, SGOT, SGPT and SBP.

Discussion (line 356)

Although our study has suggested a possible causal role of UA in the CVD, however, the biological mechanism is still not clear. Several possible mechanisms have been proposed to explain the relationship between UA and CVD development. …………………Although our MR study did not find association between SUA genetic score and markers of liver function (Table 2-1), the interrelationship of UA, NAFLD and CVD is worthy of further in-depth investigation

Table 2-1 and supplementary Table S4

SGOT and SGPT

Both NAFLD and SUA exhibit a definite sexual dimorphism. Comment on this.

Responses:

Thank you for your suggestion. We have added a paragraph to discuss this issues.

Discussion (line 385)

…… Besides, previous studies [56] have shown that the prevalence and severity of NAFLD are higher in men than in women in young adulthood and a crossover phenomenon between sexes was seen after menopause [57]. These sexual dimorphism phenomena in both traits may suggest a link between pathogenesis of hyperuricemia and NAFLD….

This study uses sophisticated and somewhat uncommon statistical techniques. Make sure that these techniques are fully explained in clear terms so as to be rendered fully accessible also to clinicians who are inexperienced in this area. In particular, describe strengths and limitations of these statistical techniques.

Responses:

To motivate interested clinicians in getting familiar with the applied statistical techniques, we modify the following sentences:

Materials and Methods (line 107) 

Left truncation, entailed by different study entry ages of the recruited disease-free participants, is a pertinent issue in longitudinal studies with repeated measurements.

Materials and Methods (line 117)

participants for each disease (see the Supplementary Materials for details). These sophisticated novel statistical methods and packages [10, 19] have been advocated to enhance the analysis and interpretation of longitudinal studies in epidemiology and medicine.

References:

Holmes MV, Ala-Korpela M, Smith GD: Mendelian randomization in cardiometabolic disease: challenges in evaluating causality. Nat Rev Cardiol 2017, 14(10):577-590. Sekula P, Del Greco MF, Pattaro C, Kottgen A: Mendelian Randomization as an Approach to Assess Causality Using Observational Data. J Am Soc Nephrol 2016, 27(11):3253-3265.